# Influences of Composite Electrodeposition Parameters on the Properties of Ni-Doped Co-Mn Composite Spinel Coatings

**DOI:** 10.3390/ma17051200

**Published:** 2024-03-05

**Authors:** Wei Tong, Weiqiang Wang, Xiayu Leng, Jianli Song

**Affiliations:** School of Instrument Science and Opto-Electronic Engineering, Beijing Information Science and Technology University, Beijing 100192, China; 15652612399@163.com (W.T.); 13163378677@163.com (W.W.); 15881141494@163.com (X.L.)

**Keywords:** FSS interconnector, composite electrodeposition, Co-Mn composite spinel coating, Ni element doping, process parameter optimization

## Abstract

To enhance the comprehensive performance of solid oxide fuel cells (SOFCs) ferritic stainless steel (FSS) interconnectors, a novel approach involving composite electrodeposition and thermal conversion is proposed to prepare Ni-doped Co-Mn composite spinel protective coatings on FSS surfaces. The process involves the composite electrodeposition of a Ni-doped Co-Mn precursor coating, followed by thermal conversion to obtain the Co-Mn-Ni composite spinel coating. Crofer 22H was used as the substrate and orthogonal experiments were designed to investigate the influences of deposition solution pH, stirring rate, cathode current density, and the element content of Mn and Ni on the surface morphology and properties of the composite coatings, respectively. The characterization of the prepared coatings was conducted through macroscopic and microscopic morphology observations of the component surface, energy dispersive spectroscopy (EDS) analysis, and area specific resistance (ASR) testing, etc. Finally, the optimized composite electrodeposition parameters and the Mn-Ni content ratio in the solution were obtained. Experimental results indicated that the composite spinel coating prepared with the optimized process parameters exhibited excellent adhesion to the substrate, and the diffusion and migration of Cr element has been effectively inhibited. Compared with the substrate, the ASR of the coated components has also been decreased simultaneously, which provided an effective method for the surface modification of SOFC FSS interconnectors.

## 1. Introduction

With the rapid development of the world economy, the energy demand has been greatly increased, therefore countries around the world are committed to reform their energy industry structures actively to improve the energy efficiency, reduce pollution emissions, and seek alternative solutions to traditional energy sources [1]. As an emerging energy system of the third generation of fuel cells, solid oxide fuel cells (SOFCs) can directly convert chemical energy into electrical energy through chemical reactions. Due to the characteristics of a wide range of applicable fuels, high energy conversion efficiency, lower cost, portability, and environmentally friendly nature [2], SOFCs are considered as one of the best solutions for green energy in the 21st century and have vast development prospects.

As a crucial component of the solid oxide fuel cell stack, the interconnector plays the role of connecting and supporting the electrodes, distributing and isolating the fuel and oxidant gases, transmission electronics, and so on. As the working temperature of SOFC changed from high temperatures above 1000 °C to intermediate temperatures of 600~800 °C, metal interconnectors gradually replaced ceramic interconnectors due to their better conductivity, ease of manufacture, and lower cost. Additionally, among these, ferritic stainless steel (FSS) is currently considered as the most ideal material for extensive application [3]. However, under long-term exposure to a medium- to high-temperature operation environment, the interconnectors are prone to oxidation, leading to increased resistance and the electrical performance of the cell will deteriorate. Additionally, ferritic stainless steel usually contains 16~27% of Cr, and during prolonged service, the Cr element in the alloys is liable to migrate outward [4,5]. Chromium will oxide on the surface of the interconnector and a kind of volatile Cr compound will be generated in the humid environment, ultimately depositing on the cathode of the cell, causing “Cr poisoning” and the commercialization of SOFCs has been seriously influenced [6,7], as illustrated in Formulas (1)–(3).
2Cr_2_O_3_ + 3O_2_ = 4CrO_3_(1)
2Cr_2_O_3_ + 3O_2_ + 4H_2_O = 4CrO_2_(OH)_2_(2)
Cr_2_O_3_ + O_2_ + H_2_O = 2CrO_2_OH(3)

To overcome the above defects, protective coatings are often prepared on the surface of the FSS substrate to prevent the “Cr poisoning” of the cathodic, as well as improve the oxidation resistance and the conductivity of the SOFC. Among the protective coatings, Co-Mn spinel coatings have been paid more attention attributed to their high density, simple prepare methods and good electrical property et al. [8,9].

In recent years, researchers have tried various methods for the modification and optimization of Co-Mn spinel coatings to further enhance their comprehensive performances [10,11,12,13]. Among these, one significant approach is to add active metal elements or their oxides into the original Co-Mn spinel coatings. For example, doping appropriate amounts of Cu, Ni, and other metal elements to improve the adhesion performance to the substrate, conductivity, and service stability of the coating [14,15,16,17]. Besides the composition modification of the coating, the preparation process also has an important impact on the coating’s performance. The preparation methods of the Co-Mn spinel coatings mainly include magnetron sputtering, screen printing, sol–gel, and electrodeposition. [18]. In the electrodeposition process, not only is the equipment cheap and simple, but the coatings prepared are also dense and uniform. However, in traditional electrodeposition methods, element doping is often realized by a layered deposition method, and challenges related to potential differences must be considered when doping multiple elements, the process is very complex and excessively time-consuming, and the uniformity of the coating will also be affected [19].

To overcome the aforementioned issues, a composite electrodeposition method to prepare Ni-doped Co-Mn spinel coatings on the surface of Crofer 22H with a short process procedure and low cost has been proposed. Orthogonal experiments were designed for the optimization of process parameters and Mn and Ni content. The influences of the composite electrodeposition process parameters and major element content on the performance of the coating have been measured and analyzed.

## 2. Experimental Materials and Methods

### 2.1. Material and Pre-Treatment of the Substrate

The chemical composition of the substrate Crofer 22H (VDM Metals International GmbH, Werdohl, Germany) is shown in Table 1, material particles and chemical reagents are from Aladdin and Macklin, Shanghai, China. The FSS sheets were cut into samples with a dimension of 20 mm × 15 mm × 2 mm using laser cutting machine (CHEX laser, Shenzhen, China). The substrate was sequentially polished using sandpapers of 240#, 400#, 800#, and 1000# on the grinding and polishing machine of metallography MPD-1 (WHW LIGHT-MACHINE, Shanghai, China) to eliminate the surface oxidation and scratches. Simultaneously, the edges of the samples were rounded to effectively prevent the formation of edge effects during the electrodeposition process. Subsequently, the polished samples were ultrasonically cleaned in deionized water for 10 min in a 4.5 L F-030SD ultrasonic machine (Shenzhen Fuyang Technology Group Co., Ltd., Shenzhen, China) and then immersed in a 1 mol/L HCl solution for acid etching for 10 min to remove the impurities on the surface. Then, the substrate samples were further cleaned by ultrasonication in deionized water for another 5 min to remove the impurities adhered to the surface. To enhance the adhesion and bonding strength between the coating and the substrate, anodic corrosion pre-treatment was performed before the composite electrodeposition experiment. The solution composition for anodic corrosion was 50 g/L CoCl_2_ and 1 mol/L HCl, with a pH adjusted to maintain an acidic electrolyte. The treated samples were placed at the positive pole of the power supply, while two Co plates of the same size served as the negative poles. The current density was set at 27 mA/cm^2^, with a stirring rate of 800 rpm, and the corrosion time was 5 min. Finally, the pre-treated samples were ultrasonically cleaned in deionized water, dried, and immersed in anhydrous ethanol to prevent surface oxidation.

### 2.2. Prepare of the Electrodeposition Solution and Design of the Orthogonal Experiments

Orthogonal experimental design was used to study the influences of multiple important and representative factors on the quality of the prepared coatings at multiple levels, thus to optimize the process parameters and element contents. Representative process parameters such as the stirring rate, current density, and pH value were selected as the three factors exploring the influences of parameters on the quality of the prepared coating, and the contents of Mn_3_O_4_ and NiO were selected as the main factors exploring the influences of different element contents on the coating, three levels for each orthogonal experiment have been set up according to the pre-determined experiments, as presented in Table 2. The electrodeposition time is directly related to the thickness and conductivity of the coating, and the adhesion between the coating and the substrate. In the experiment, the deposition time was set in the category of 10–20 min, and after a comprehensive comparison, it was found that an electrodeposition time of 15 min is the most suitable; therefore, the electrodeposition time was selected as 15 min in the whole experiment.

Due to the presence of the physical interface between the particles and the dispersed dielectric in the prepared solution, the addition of sodium dodecyl sulfate (SDS) and the complexing agent EDTA-2Na was employed to enhance the stability of the electrolyte and effectively prevent the aggregation of hydrophobic particles in the solution. A 300 mL electrodeposition solution was prepared using deionized water, and ultrasonic and magnetic stirring was adopted to ensure the uniform dispersion of particles in the solution. The composition of the electrodeposition solution is listed in Table 3. The composition of the electrolyte was initially determined through pre-experiments. After the optimization of the process parameters, another set of orthogonal experiments was conducted to determine the optimal content of Mn and Ni elements in the electrolyte to explore the influences of element content on the conductivity of the spinel-coated samples.

### 2.3. Preparation Method for the Ni-Doped Co-Mn Spinel Coating

Ni-doped Co-Mn composite precursor coating was prepared using a composite electrodeposition method, followed by thermal conversion to obtain the Co-Mn-Ni composite spinel coating. Initially, a pretreated Crofer 22H sample was installed at the cathode, and two Co plates with the same size were taken as the double anodes to ensure better deposition quality. The electrode space was set at 35 mm, the total volume of the deposition solution was 300 mL, and the composite electrodeposition time was 15 min. The composite precursor coating prepared by the composite electrodeposition was then heated to 800 °C at a rate of 1 °C per minute and held for one hour to transform the precursor coating into a more stable composite spinel coating. The principle of the Ni-doped Co-Mn composite spinel coating preparing process is illustrated in Figure 1.

### 2.4. Performance Testing and Characterization Methods of the Composite Coatings

After the preparation of the composite precursor coating and spinel coating, the surface morphology of the Ni-doped Co-Mn composite precursor coating and spinel coating was observed using a Hitachi cold field emission scanning electron microscopy (SEM) Regulus 8100 (HITACHI, Tokyo, Japan). Additionally, energy-dispersive spectroscopy (EDS) composition analysis was conducted using the attached energy spectrum analyzer to determine the distribution of elements such as Mn and Ni within the coatings and investigate the diffusion of elements in the cross-section. For the spinel-coated samples prepared under the optimized parameters with the orthogonal experiments, area-specific resistance ASR testing was carried out using a direct current four-point probe method in a box-type resistance furnace at temperatures ranging from 600 to 800 °C in an air environment. The influences of Mn and Ni in the solution on the electrical conductivity of the spinel coating were studied during the ASR measurement.

## 3. Results and Discussion

### 3.1. Influence Mechanisms of the Process Parameters on the Composite Coatings

The morphologies of the samples prepared through the orthogonal experiments for process parameter optimization were analyzed by macro and microscopic observation. The results indicated that coatings with better comprehensive quality were obtained under the conditions of a current density of 45 mA/cm^2^, pH value of 4.0, and a stirring rate of 600 r/min. Macro- and microscopic morphologies of the composite precursor coating and spinel coating are shown in Figure 2. Macroscopically, both the precursor coating and the spinel coating after oxidation exhibited smooth surfaces with complete coverage over the substrate, as illustrated in Figure 2a,c. The microscopical SEM observation showed that the oxide metal particles in the precursor coating were uniformly distributed, as seen in Figure 2b, while the grains of the spinel coating surface were dense and basically uniform without any observed pores, peeling or other defects, as shown in Figure 2d. The specific analysis of the influence mechanisms of the key process parameters on the composite electrodeposition process and the formation of the spinel coating will be analyzed below in detail.

#### 3.1.1. Influence of the pH Value

At a constant current density of 45 mA/cm^2^ and a stirring rate of 600 r/min, the surface morphology of the spinel coatings—prepared under different pH values (4.0, 5.0, and 6.0) were shown in Figure 3, the left side was the morphologies of the prepared spinel coatings, and the right side was the distribution of the Mn element in the spinel coatings. From the images, it was evident that, at pH5, protrusions in some areas of the coating surface have been generated, and at a higher magnification of SEM, numerous pores can be observed. At pH6, the coating surface was highly irregular, and the distribution of the spinel structure was extremely uneven, which was attributed to the influence of the solution pH on the efficiency of the current, and thereby the quality of the coatings. The deposition effect of the coating was closely related to the pH value of the solution. When the pH value of the deposition solution was greater than the isoelectric point (pI) of the deposited particles, the selected material would carry a negative charge. The higher the pH value, the more negative charges would be carried, resulting in a negative Zeta potential of the solution. Only when the material particles are positively charged can they be effectively absorbed and planted into the coatings of the cathodic substrate. As the pI value of Mn_3_O_4_ is 5.7 and the pI of NiO is about 9.9–11.3, when pH = 6, it became challenging to deposit oxide particles especially Mn_3_O_4_ into the coatings of the substrate through the adsorption effect of the electric field, which was the main component that constituted the Co Mn spinel structure. Instead, the deposition was primarily driven by the kinetic energy of stirring during the deposition process, causing the uneven distribution of metal oxide particles on the coating surface, as shown in the right image of Figure 3c. Even at pH5, where the pH value of the deposition solution was a little lower than the pI value of the deposited particles, the relatively small potential difference still resulted in the weak adsorption of metal particles toward the cathodic substrate. In this case, the deposition process was still dominated by the kinetic energy of stirring, leading to the appearance of some protrusions on the surface of the samples, as shown in the left image of Figure 3b. In contrast, at pH4, the material particles in the solution carried a positive charge. During the electrodeposition process, the nanoparticles of Mn_3_O_4_ and NiO can be efficiently implanted into the Co matrix with the growth of the coating. The prepared coating surface, when observed under the high magnification of SEM, was dense and uniform, exhibiting good surface quality, as shown in Figure 3a.

#### 3.1.2. Influences of the Current Density and Stirring Rate on the Spinel Coating

As mentioned above, when the pH value was lower than the isoelectric point (pI) of the deposited particles, the material particles exhibited a positive charge. In this section, the deposition effect on the cathodic substrate is mainly determined by the kinetic energy of stirring and the potential difference between the material particles and the cathode. pH4 was selected for the analysis of the influences of the stirring rate and cathodic current density on the deposition process.

Figure 4, Figure 5 and Figure 6 showed the SEM morphologies of the composite electrodeposition thermal conversion spinel coatings prepared at different stirring rates under three current densities of 25 mA/cm^2^, 35 mA/cm^2^, and 45 mA/cm^2^, respectively. It can be observed that, when the current density was low, the deposition effect of the coating was not ideal under all three stirring rates. This is because, at low current densities, the potential difference between the cathode and the particles in the solution was lower than the minimum threshold required for the composite electrodeposition. Consequently, the deposition efficiency of metal particles in the deposition solution was significantly reduced. It primarily relied on the kinetic energy of stirring to slowly implant along with the reduction of Co ions at the cathode in the composite precursor coating. However, because the current density is too low, the growth efficiency of the coating was very low correspondingly. Therefore, the deposition effect of solid oxide particles was poor, as shown in Figure 4. When the current density was at the medium to high level, the deposition of Co at the cathode was effective due to a sufficient potential difference. Simultaneously, the random effects caused by the stirring process can be partially offset by the adsorption of particles from the solution onto the cathodic substrate. By comparing the microstructures, it can be noticed that, at medium to high stirring rates, the coating still exhibited an uneven distribution of materials due to the significant flushing effects. Under high magnification, some sample surfaces even showed defects such as pores and dents, as demonstrated in Figure 5c. However, at a stirring rate of 600 r/min, spinel coatings prepared under medium to high current density conditions were relatively uniform and smooth. When the current density was 35 mA/cm^2^, there were protrusions in some areas of the coating, as shown in Figure 5a. At a current density of 45 mA/cm^2^, the spinel structure was evenly distributed on the coating surface, resulting in optimal overall coating performances, as shown in Figure 6a.

### 3.2. Influences of Mn_3_O_4_ and NiO Content on the Coatings

#### 3.2.1. Influences of Mn_3_O_4_ Content on the Coatings

Manganese (Mn) exhibits excellent conductivity at high temperatures, and its thermal expansion coefficient is close to that of Crofer 22H. The amount of Mn in the coating is crucial for the stability and conductivity of the spinel coating. Figure 7 shows the SEM morphologies of the precursor coatings prepared under different Mn_3_O_4_ contents. It can be observed that, when the Mn_3_O_4_ content was 133 g/L, the distribution of oxide on the coating surface was not uniform, and the implantation into the Co matrix was relatively low. When the Mn_3_O_4_ content was 167 g/L, the coating presented good density and uniformity. However, when the Mn_3_O_4_ content was increased to 200 g/L, Mn_3_O_4_ elements tended to aggregate in local areas (marked in a red circle) due to excessively high Mn_3_O_4_ content in the solution, leading to uneven distribution. Figure 8 illustrates the element content of Mn and Ni in the composite electrodeposited precursor coating at different Mn_3_O_4_ contents in the deposition solution. This showed that, when the Mn_3_O_4_ content was in the range of 133 g/L–167 g/L, the Mn content in the coating increased with the addition of Mn_3_O_4_ to the solution. However, when Mn_3_O_4_ was continuously added into the deposition solution, the Mn content in the coating tended to become saturated. Even with an increase in Mn_3_O_4_ concentration in the solution, the increase in Mn content in the coating was not significant and the implantation of Ni into the Co matrix would even be affected. Therefore, the most suitable Mn_3_O_4_ addition to the deposition solution was 167 g/L.

#### 3.2.2. Influences of NiO Content on the Coating

Nickel (Ni) exhibits excellent performance under high temperatures, with the ability to withstand temperatures up to 1200 °C. Additionally, Ni has a good affinity with the substrate, which can increase the adhesion between the coating and the substrate. In some extreme environments, Ni also demonstrates corrosion resistance. Therefore, a dense spinel coating suppressing the outward migration of chromium (Cr) from the substrate and cathode “Cr poisoning” can be generated with the introduction of metallic Ni into Co–Mn spinel coatings. Furthermore, Ni doping will promote the formation of the spinel phase, stabilizing the crystal structure of the coating, and thereby enhancing the thermal and chemical stability of the sample. In the working environment of SOFC, the thermal expansion coefficient can be decreased by Ni doping in Co-Mn spinel coatings, surface damage and cracking induced by internal stress can be reduced and the electrical conductivity of the spinel coating can also be improved [15,16]. However, the optimal amount of Ni addition should be optimized based on different materials and processes. 

To investigate the influence of the Ni element content on the quality and electrical conductivity of Co-Mn spinel coatings, an orthogonal experiment was conducted during composite electrodeposition with a NiO content variation at different levels of 33 g/L, 67 g/L, and 100 g/L. The prepared precursor coatings were subsequently heat treated to convert into spinel coatings at 800 °C in air, and their electrical conductivity was tested. The area-specific resistances of the spinel coatings with different NiO contents at 700 °C and 800 °C are shown in Figure 9. As shown in the figure, the area-specific resistance of the spinel coatings decreased first and then increased at both temperatures. When the Mn_3_O_4_ addition was 167 g/L and the NiO addition was 67 g/L, the spinel coating exhibited the best electrical conductivity. It indicated that a certain amount of Ni element doping can enhance the electrical conductivity of the coating. However, when the NiO content reached a critical value, the electrical conductivity decreased gradually. This was because, with an increasing NiO content, the amount of Mn_3_O_4_ implanted into the precursor coating will decrease, thereby the electrical conductivity would be affected. Combining the optimization of the process parameters and Mn content, it can be summarized that, when the NiO content was 67 g/L, the prepared spinel samples had improved electrical conductivity compared to the substrate in the temperature range of 600–800 °C, as shown in Figure 10. With the increasing temperature, the ASR of the spinel coating will decrease accordingly. At 800 °C, the ASR of the Ni-doped composite Co-Mn spinel coating was 1.1 mΩ·cm^2^, which is relatively lower and the application of SOFC interconnect can be satisfied.

#### 3.2.3. Influences of NiO Doping on the Adhesion between the Coating and the Substrate and the Inhibition Performance of Cr

Figure 11 provides the cross-sectional SEM morphology of the Ni-doped Co-Mn composite precursor coatings and spinel coatings, along with the EDS line scan images. As observed in the figures, oxide particles are basically uniformly embedded into the precursor coating through composite electrodeposition, as shown in Figure 11a. The spinel coating exhibited excellent adhesion to the substrate, and the coating was relatively dense, attributing to the adding and function of element Ni. The reaction between Mn, Ni, and Cr with O_2_ within the composite coating and the substrate and elements diffusion in the interface occurred during the thermal conversion, as shown in Figure 11c,d. By a comparison of the EDS line scan images, it can be seen that due to the lack of high-temperature oxidation of the composite precursor coating, the Cr element fluctuated on the surface of the precursor coating due to the incomplete coverage of the precursor coating surface by the oxides, as shown in Figure 11b. The Cr content scanned on the surface of the spinel coating surface was extremely lower, which exhibited an excellent Cr inhabiting performance of the Ni-doped Co-Mn spinel coating, as shown in Figure 11d.

Through orthogonal experiments on the optimization of Mn and Ni element contents and the analysis of the influences of element contents on the coating, it can be concluded that, when the NiO content was 67 g/L and Mn_3_O_4_ content was 167 g/L, the prepared spinel coating exhibited a uniform element distribution. Additionally, conductivity tests revealed that the spinel coating at the optimized Mn and Ni ratio demonstrated the best electrical performance, as described in Figure 12.

## 4. Conclusions

A composite Ni-doped Co-Mn spinel coating has been successfully prepared on the surface of Crofer 22H through a composite electrodeposition method followed by thermal conversion at 800 °C for 60 min to prepare the precursor and spinel coatings, respectively. The influences of different Mn and Ni contents in the composite electrodeposition solution and process parameters on the quality and performance of the coating were optimized and analyzed through orthogonal experiments. A protective spinel coating with good conductivity, good bonding strength, high density, and surface quality has been prepared. The main conclusions can be drawn as follows:(1)The influences of the stirring speed, pH value, and cathodic current density on the composite electrodeposition process of the precursor coating were studied. Combining the SEM surface morphology observations and EDS analysis, the optimized composite electrodeposition process parameters were determined as follows: a stirring speed of 600 rpm, pH = 4, and the current density i = 45 mA/cm^2^.(2)The influences of Mn and Ni contents in the composite electrodeposition solution on the coating performances were studied. The optimized Mn_3_O_4_ content was determined to be 167 g/L, and the optimized NiO content was 67 g/L, under which the best performances of the composite coating can be obtained.(3)The ASR of the Ni-doped Co-Mn composite spinel coating was measured at 800 °C using the four-probe method. The ASR of the spinel coating was 1.1 mΩ·cm^2^. Compared to the ASR of the bare substrate of 5.58 mΩ·cm^2^, the conductivity of the coated substrate was significantly improved.(4)The composite spinel coating prepared with the optimized process parameters and deposition solution composition exhibited excellent adhesion to the substrate, uniform composition distribution, and good resistance to Cr diffusion. The oxidation resistance of the substrate can be further improved contributing to the dense spinel coating structures and good adhesion to the spinel coatings.(5)Further studies will be conducted on the oxidation behavior and conductive performance of the spinel coatings under long-term oxidation conditions. Additionally, comparative experiments will be carried out with the coating performances prepared with other methods.

## Figures and Tables

**Figure 1 materials-17-01200-f001:**
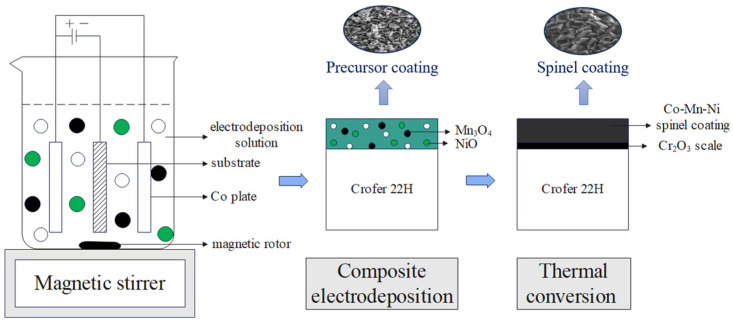
Schematic diagram of the preparation process for the composite spinel coating.

**Figure 2 materials-17-01200-f002:**
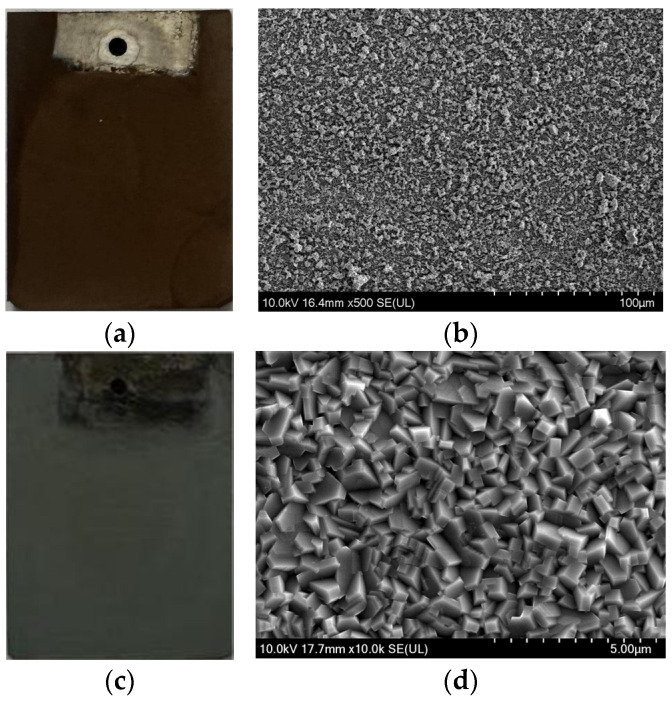
(**a**) Macroscopic morphology of the composite precursor coating of the sample; (**b**) SEM morphology of the precursor coating; (**c**) macroscopic morphology of the composite spinel coating of the sample; and (**d**) SEM morphology of the spinel coating.

**Figure 3 materials-17-01200-f003:**
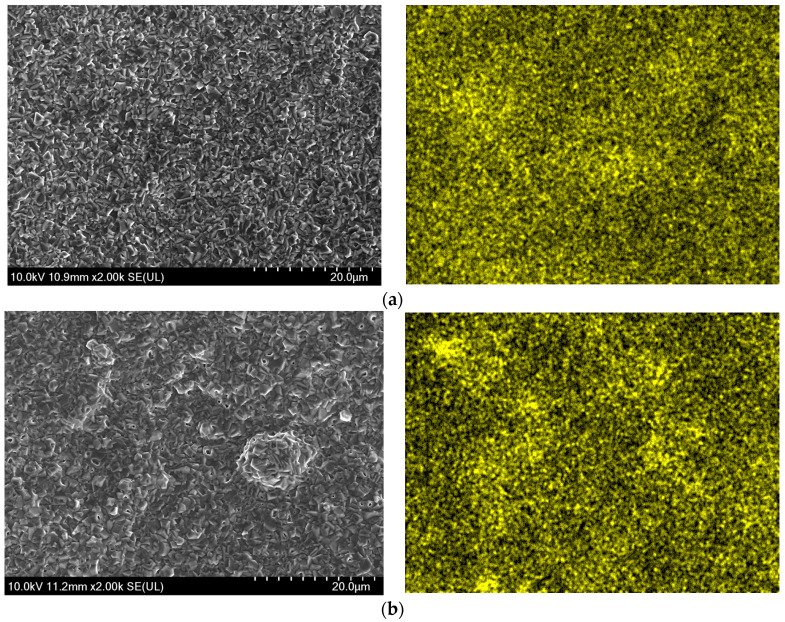
SEM morphologies (**left**) and Mn element EDS distribution (**right**) on the spinel coating surface: (**a**) at pH4; (**b**) at pH5; and (**c**) at pH6.

**Figure 4 materials-17-01200-f004:**
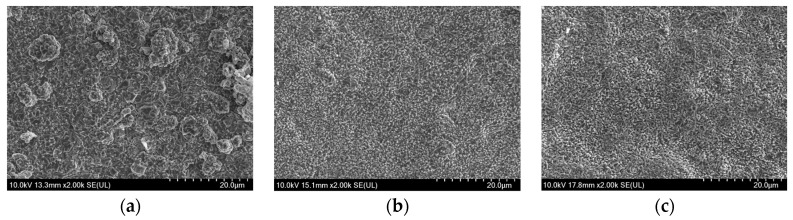
SEM morphologies of the spinel coating surface under the current density of 25 mA/cm^2^ and at a stirring rate of (**a**) 600 rpm, (**b**) 700 rpm, and (**c**) 800 rpm.

**Figure 5 materials-17-01200-f005:**
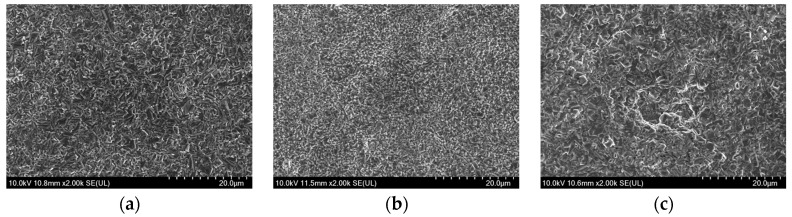
SEM morphologies of the spinel coating surface under the current density of 35 mA/cm^2^ and at a stirring rate of (**a**) 600 rpm, (**b**) 700 rpm, and (**c**) 800 rpm.

**Figure 6 materials-17-01200-f006:**
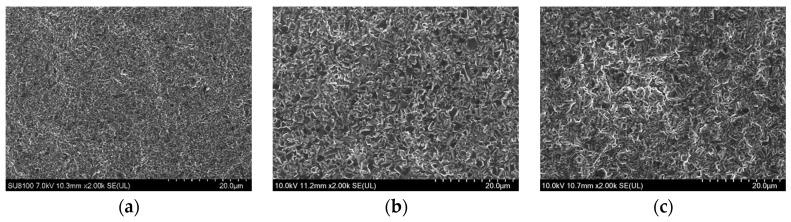
SEM morphologies of the spinel coating surface under the current density of 45 mA/cm^2^ and at a stirring rate of (**a**) 600 rpm, (**b**) 700 rpm, and (**c**) 800 rpm.

**Figure 7 materials-17-01200-f007:**

SEM morphology of the precursor coating surface at a Mn_3_O_4_ content of (**a**) 133 g/L, (**b**) 167 g/L, and (**c**) 200 g/L (Oxides tended to aggregate in red circles).

**Figure 8 materials-17-01200-f008:**
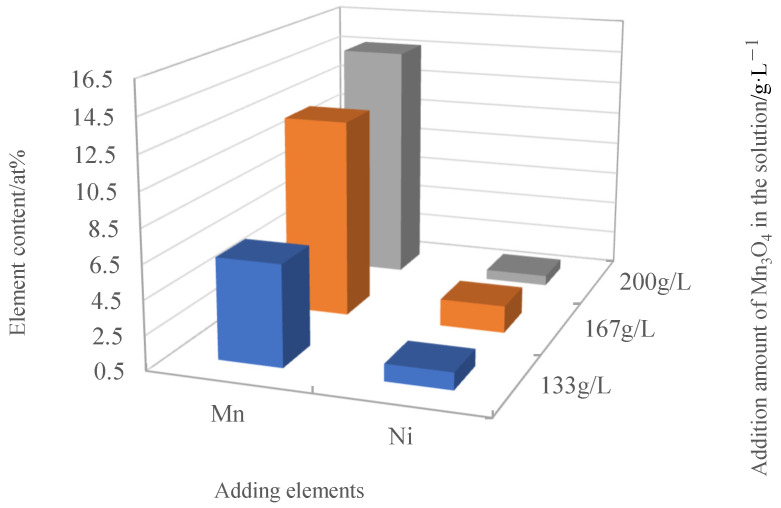
Mn and Ni content in the precursor coating.

**Figure 9 materials-17-01200-f009:**
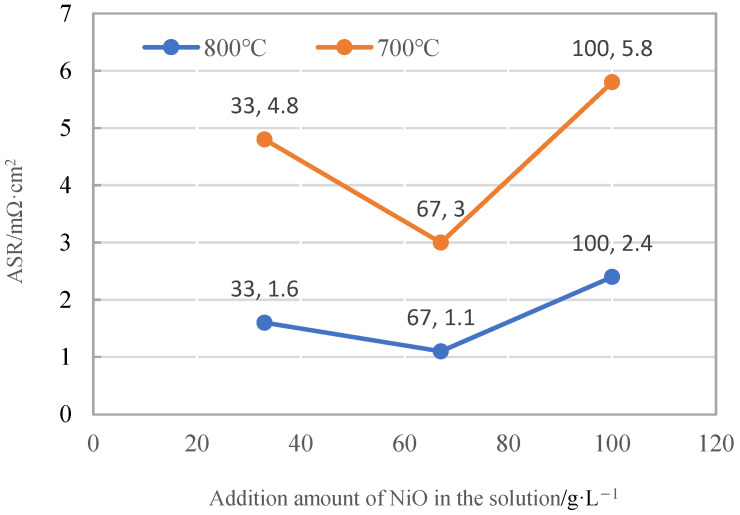
ASR curves for the spinel coatings varied with different contents of NiO in the solution.

**Figure 10 materials-17-01200-f010:**
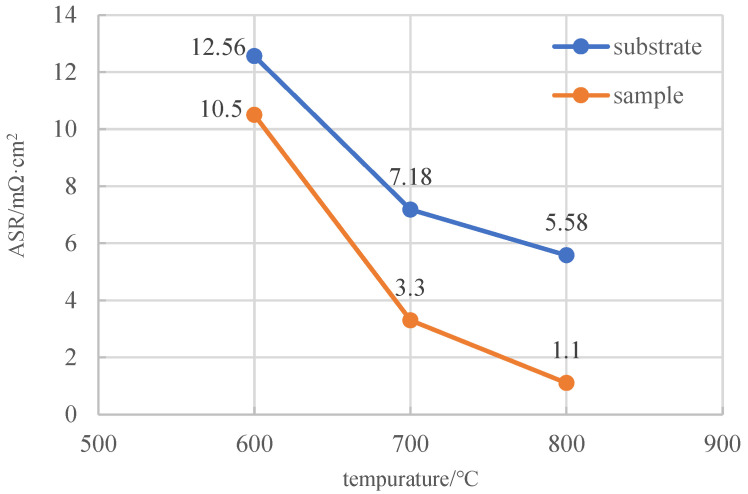
ASR curves of the spinel samples and bare substrate varying in the range of 600–800 °C.

**Figure 11 materials-17-01200-f011:**
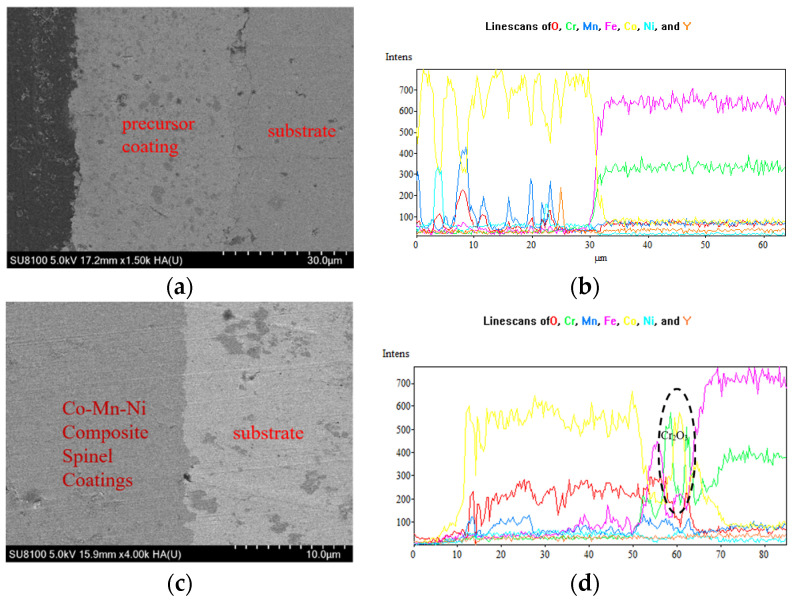
(**a**) Cross-sectional morphology of the precursor coating; (**b**) EDS line scan of the precursor coating; (**c**) cross-sectional morphology of the spinel coating; and (**d**) EDS line scan of the spinel coating.

**Figure 12 materials-17-01200-f012:**
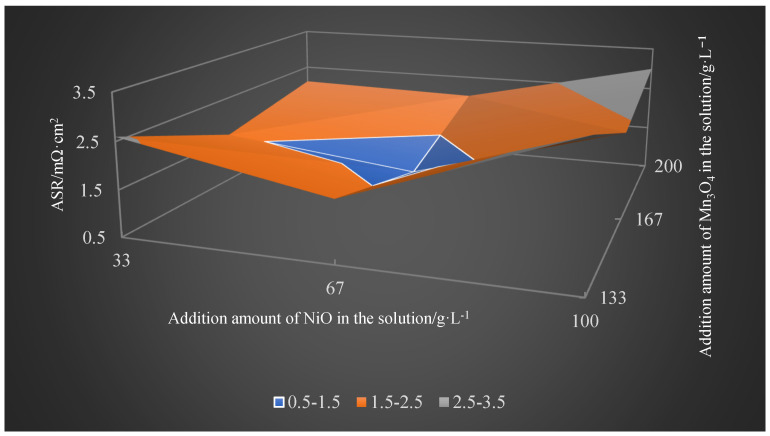
ASR of the spinel samples with different Mn_3_O_4_ and NiO additions at 800 °C.

**Table 1 materials-17-01200-t001:** Chemical composition of the Crofer 22H substrate (wt.%).

Chemical Composition	Cr	C	N	S	Mn	Si	Al	W	Nb	Ti	La	P	Ni	Cu	Fe
wt. (%)	23.0	0.01	0.02	<0.002	0.44	0.25	0.01	1.9	0.50	0.10	0.07	0.016	0.33	0.02	Bal.

**Table 2 materials-17-01200-t002:** Setting of the experimental conditions and parameters for composite electrodeposition.

Design of the Experiments	Factors	1	2	3
(1—Orthogonal experiment for optimization of process parameters): to explore the influences of different process parameters on the prepared coatings	Stirring rate (rpm)	600	700	800
	Current density (mA/cm^2^)	25	35	45
	pH Value	4	5	6
(2—Orthogonal experiment for the optimization of the main adding element contents): To explore the influences of different element contents on the coating	Mn_3_O_4_ g/L	133	167	200
	NiO g/L	33	67	100

**Table 3 materials-17-01200-t003:** Composition of the composite electrodeposition solution.

Reagents	Concentration
NiO	33/67/100 g/L
CoCl_2_	50 g/L
CoSO_4_·5H_2_O	250 g/L
H_3_BO_3_	30 g/L
Y_2_O_3_	10 g/L
SDS	0.05 g/L
EDTA-2Na	20 g/L
Mn_3_O_4_	133/167/200 g/L

## Data Availability

The data that support the findings of this study are available from the corresponding author, Song, upon reasonable request (due to privacy).

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
