# Peer review of "Influences of Composite Electrodeposition Parameters on the Properties of Ni-Doped Co-Mn Composite Spinel Coatings"

_materials, 2024, doi:10.3390/ma17051200_

Round 1

Reviewer 1 Report

Comments and Suggestions for Authors

This manuscript presents a study on the influences of composite electrodeposition parameters on the properties of Ni-doped Co-Mn composite spinel coatings. The authors performed a series of investigation on optimizing the conditions to deposit Ni- doped Co-Mn spinel coatings with superior properties. The article is fairly well written and relevant in the field. The results presented are interesting, however needs a bit more clarity to be published in ‘coatings’. Therefore, this article could be considered for publication in ‘coatings’ after addressing the comments below.

1.     Full form should be used when used in the first time in the article, SOFC in the abstract for example.

2.     Authors need to explain what they mean by orthogonal experiments. It may not be clear for a non-expert reader if not well explained. One or two sentences in the experimental part would be OK.

3.     Line 50, “a kind of a volatile Cr compound”. Can you be more specific here? With example from literature.

4.     The sentence from lines 52-55 is not clear, there are other sentences and wordings also in the manuscript which lacks clarity. Authors should rewrite and make sure they are clear.

5.     Lines 63-64, “The preparation methods……” needs appropriate references here.

6.     Lines 86-87, “ultrasonication … for another time”. Need to be specific.

7.     Why was the electrodeposition time set to 15 min.? Any relation between thickness of material and the properties?

8.     Figure 2 low magnification image, the authors wrote ‘smooth surfaces’ based on the image. But it doesn’t look very smooth from the image. Please explain.

9.     I wonder whether the authors tried any lower pH values (than 4) and see how the deposit morphology changes?

10.Figure 4, The discussion on deposition efficiency is based on three top-view SEM images. Seems not much difference between 4b and 4c. Are there any differences in thickness, roughness, composition etc.?

11.Figure 5C is not very high magnification image. So, its difficult to see whether it contains defects and pores or dents as the authors explain. Is there any high magnification image to show it clearly?

12. The authors discuss the difference in potential between different current density. Can you show how it was measured, by showing a representative t vs E plot for example.

13. Line 231, “stirring rate was 235 mA/cm2” correct the error.

14.Figure 7, the authors say good smoothness, but not obvious. Also, it is difficult to conclude from the images that the 200 g/L leads to aggregation and fluctuating surface morphologies.

15.I wonder whether it is metallic Ni or the oxide is present in the film. Lacks clarity on it.

Comments on the Quality of English Language

English language needs bit of improvement and polishing of sentences required. 

Author Response

Dear Reviewer,

Thank you for your detailed review and insightful comments on our manuscript. We have carefully considered each of your comments and made the following revisions and clarifications:

Comment 1: Full form should be used when used in the first time in the article, SOFC in the abstract for example.

Response: We have ensured that the first mention of abbreviations, such as "SOFC," is written in full form, as requested.

Comment 2: Authors need to explain what they mean by orthogonal experiments. It may not be clear for a non-expert reader if not well explained. One or two sentences in the experimental part would be OK.

Response: The method of orthogonal experimental design in the manuscript has been explained briefly in the experimental section to enhance clarity for non-specialist readers.

Comment 3: Line 50, “a kind of a volatile Cr compound”. Can you be more specific here? With example from literature.

Response: We have provided a more specific explanation regarding the volatile Cr compound mentioned in line 50, drawing from relevant literature, and added three formulas to illustrate the process, as shown in the follows:

2Cr2O3+3O2=4CrO3                                                             (1)

2Cr2O3+3O2+4H2O=4CrO2(OH)2                                                 (2)

Cr2O3+O2+H2O=2CrO2OH                                                      (3)

Comment 4:  The sentence from lines 52-55 is not clear, there are other sentences and wordings also in the manuscript which lacks clarity. Authors should rewrite and make sure they are clear.

Response: We feel very sorry for the grammar mistakes and have already corrected them to make the sentences more clearly expressed. Thank you for your kind reminder.

Comment 5: Lines 63-64, “The preparation methods……” needs appropriate references here.

Response: We have added appropriate references to support the preparation methods mentioned in lines 63-64.

Comment 6: Lines 86-87, “ultrasonication … for another time”. Need to be specific.

Response: Beforehand, we subjected the samples to acid cleaning, wherein impurities on the sample surface were further dissolved under the acidic conditions. However, some impurities might have adhered to the surface. Ultrasonic cleaning can effectively remove surface contaminants while also facilitating better retention of acidic solution on the substrate surface. We have already specified the corresponding part.

Comment 7: Why was the electrodeposition time set to 15 min.? Any relation between thickness of material and the properties?

Response: The electrodeposition time is directly related to the thickness of the coating, conductivity and adhesion between the coating and the substrate. If the coating is too thin, its ability to protect the sample from high-temperature oxidation and corrosion of Cr will be reduced, and it is prone to crack and pores are easy to generate during operation. On the other hand, if the coating is too thick, the conductivity of the interconnector will decrease, and the adhesion to the substrate will also be weakened. Therefore, it is necessary to adjust the deposition time to ensure that the coating thickness falls within an appropriate range. In the experiment, the deposition time was set in the range 0f 10 to 20 minutes, and after comprehensive comparison, it was found that an electrodeposition time of 15 minutes is the most suitable. As it falls outside the scope of orthogonal experimental design, it was not detailly described in this article and will be write in another manuscript. In the revised version, it is briefly supplemented by the authors according to the reviewer’s comment.

Comment 8: Figure 2 low magnification image, the authors wrote ‘smooth surfaces’ based on the image. But it doesn’t look very smooth from the image. Please explain.

Response: The term "smooth surface" primarily describes the macroscopic state of the substrate, such as figure 2 (c) in the low magnification image of figure 2. In the middle and upper parts of the component, the fixture clamps were applied during the experiment, and there was no any spinel coating in the area. The coating is evenly distributed without significant irregularities or peeling. Therefore, it is “smooth” on the surface of the prepared coating.

Comment 9: I wonder whether the authors tried any lower pH values (than 4) and see how the deposit morphology changes?

Response: In fact, we previously attempted to conduct experiments with pH levels below 4, but the results were not satisfactory. Due to the excessively acidic environment, significant peeling of the deposited coating occurred. Therefore, the pH level in the orthogonal experiments was selected from pH4 according to the results of pre-determined experiments, and other parameters are also investigated, not limited to those listed in the text.

Comment 10: Figure 4, The discussion on deposition efficiency is based on three top-view SEM images. Seems not much difference between 4b and 4c. Are there any differences in thickness, roughness, composition etc.?

Response: Between 4b and 4c, besides the difference in the rotation speed, all the other parameters remained consistent. Due to the lower current density of 25 mA/cm² and varying of the rotation speeds, both 4b and 4c exhibited uneven distribution of the composite particles on coating surface and rough surface, and thickness of the coating should be varied gently under different stirring speed, but the thickness and composite distribution of the coating have not been studied under a result of non-optimized surface quality.

Comment 11: Figure 5C is not very high magnification image. So, it’s difficult to see whether it contains defects and pores or dents as the authors explain. Is there any high magnification image to show it clearly?

Response: Because the three pictures should be in the same magnification, and we have check the picture of figure 5c, it can be seen that obvious defects can be observed in the middle of figure 5c.

Comment 12: The authors discuss the difference in potential between different current density. Can you show how it was measured, by showing a representative t vs E plot for example.

Response: The Isoelectric point (pI) refers to the pH value at which a molecule in a certain solution exhibit neither surface charge nor net charge. That is, in a given solution at a certain pH value, when the positive and negative charges of a molecule are exactly balanced (resulting in a net charge of zero), it will not move towards either the anode or cathode in an electric field. When the pH value of the deposited solution is less than the pI value of the deposited particle, the selected material will be positively charged, and a potential difference will be formed between the selected material and the substrate connected to the cathode. The lower of the pH than the pI, the more positive charge will be carried by the particles, and the greater the default potential difference will be gained.

Comment 13: Line 231, “stirring rate was 235 mA/cm2” correct the error.

Response: Thank you for pointing out this problem in our manuscript. The error in line 231 has been already corrected.

Comment 14: Figure 7, the authors say good smoothness, but not obvious. Also, it is difficult to conclude from the images that the 200g/L leads to aggregation and fluctuating surface morphologies.

Response: Here “smooth” is misused in this paragraph,it can only observed from macrograph image, we have revised it in the text. The aggregation and nonuniform areas in the image of 200 g/L has been marked as follows. Aggregates are showed in the red circles, while the areas in green circles are not covered by the oxide. Therefore, the coating is nonuniform.

Comment 15: I wonder whether it is metallic Ni or the oxide is present in the film. Lacks clarity on it.

Response: EDS analysis on the coating surface and cross-section with SEM attachment showed that the coatings do contain nickel oxides as illustrated in Figure 11 in the text. Additionally, the material used in this experiment is NiO so the nickel is in the form of nickel oxide.

We appreciate your thorough evaluation of our manuscript and believe that these revisions will significantly enhance the quality and clarity of our work. We have provided necessary revisions and clarification regarding your comments and questions. Thank you again for your valuable comments, please let us know if further clarification or revisions are needed.

Best regards,

The authors

26 Feb.,2024

Reviewer 2 Report

Comments and Suggestions for Authors

Congratulations. The article shows good knowledge by the authors in the area. However, some doubts about your work can be solved by improving the work and their understanding.

Author Response

Dear Reviewer,

Thank you for your thorough review of our manuscript and for providing valuable suggestions for improvement our manuscript. We appreciate your attention to the details and are committed to addressing each of your points effectively. Here are the responses to your comments:

Comment1. Line 40/49, in the introduction: The authors make some statements without being supported by references. Authors must add references to validate all statements.

Response: The statements in lines 40/49 of the Introduction have been revised, and appropriate refences have been added to support the statements.

Comment2. Line 53, the authors use the abbreviation FSS for the first time in the text. Even though they have written it in full in the summary, they must also do so when they mention it for the first time in the text. Check whether there are more identical situations with other abbreviations. Further ahead in the work, ASR…

Response: We have checked the full text, and revised abbreviations have not written in full in the first time in the text, such as “FSS”.

Comment3. Line 80, in chapter 2 - Experimental Materials and Methods, the authors state that they used a laser cutting method. It remains to present the equipment used in laser cutting.

Response:

The equipment used in laser cutting in Line 80 has been added in the manuscript.

Comment4. Lines 80/81. The authors are not clear on the polishing process. Was any equipment used or was everything done manually? If you used the equipment, they must mention it.

Response: Regarding the polishing process mentioned in lines 80/81, we apologize for any confusion. The surface polishing was manually conducted on the grinding and polishing machine of metallography with sandmanuscript, and the type of the machine has already been added in the manuscript.

窗体顶端

Comment5. In ultrasonic cleaning, what was the concentration/quantity used in the process? It remains to add the ultrasound equipment used.

Response: In ultrasonic cleaning, the ultrasonic equipment used has been added in the revised manuscript and the solution used is deionized water as stated in the manuscript.

Comment6. - It would be interesting if the authors presented one or two images of the equipment used in the study and presented in sub-chapter 2.1. Material and Substrate Pretreatment.

Response: The pre-treatment equipment, is indeed the same as that used in the formal experiments, as described in Figure1. We think that there is no great significance to present the images of the pre-treatment equipment in the manuscript.

Comment7. What was the number of samples used in this study?

Response: The number of the sample utilized in our study 36 groups, as clarified in the section of orthogonal experiments.

Comment8. - Line 312. The numbering is wrong. It has 2.2.3 and should have 3.2.3.

Response: We have corrected the numbering error in line 312 to 3.2.3, thank you.

Comment9. - Figure 1 can be enlarged a little to better read the captions in the image on the left.

Response: We have redrawn Figure 1.

Comment10. Figures 8, 9 and 10 may have the same dimension. This makes it more uniform.

Response: We have revised the dimension of Figures 8, 9, and 10 according to the comments, thank you.

Comment11. In figure 11, there are (a) and (b) twice. They should switch to (c) and (d). They must also be placed with the same dimension. (b) and (d) must be aligned, since they are the same gender. (d) has the word “Intens” cut off on the YY axis.

Response: We have revised figure11 according to the comments.

Comment12. It would be interesting for the authors to compare their results with the results of other authors.

Response: Thank you for your thoughtful comment. We appreciate your suggestion regarding comparing our results with those of other authors. However, up to now, we have not indexed any spinel coatings prepared with the same experimental methods and substrate materials by others authors. For example, in reference 16, the substrate used is ss430, and the spinel coating is prepared layer by layer. Compared with the method in reference 16, the procedure has been shortened, and compared with the bare sample without Ni adopting, the conductivity has been greatly improved. And other comparative experiments are scheduled to be conducted later under the same experimental conditions.

Comment13. - The conclusions are objective, factual and in accordance with the results presented, however, there is a lack of criticism based on the scientific experience acquired throughout the work. When they state that they obtained a certain result, they must add whether it meets expectations and existing literature, whenever possible.

Response: From the experiment and research results,we tried to development an approach to prepare a protective coating with good comprehensive performance, low cost and short procedure. The results is relatively satisfied according to the comparation with the substrate, but there still many other works to do in the future, and comparation will be meaningful only with the same substrate and test conditions. We have added the works will be further done in the future in the conclusion. And comparative experiments maybe done by another student. Thank you again for you good suggestion.

Thank you once again for your insightful feedback, and we are dedicated to implementing these revisions to enhance the clarity and quality of our manuscript.

Best regards,

The authors

26 Feb.,2024

Round 2

Reviewer 1 Report

Comments and Suggestions for Authors

Thank you for attending the comments. This manuscript could be accepted for publication